# Acacetin Inhibits the Growth of STAT3-Activated DU145 Prostate Cancer Cells by Directly Binding to Signal Transducer and Activator of Transcription 3 (STAT3)

**DOI:** 10.3390/molecules26206204

**Published:** 2021-10-14

**Authors:** Sun Yun, Yu-Jin Lee, Jiyeon Choi, Nam Doo Kim, Dong Cho Han, Byoung-Mog Kwon

**Affiliations:** 1Laboratory of Chemical Biology and Genomics, Korea Research Institute of Bioscience and Biotechnology, 125 Gwahakro, Daejeon 34141, Korea; yunsun412@kribb.re.kr (S.Y.); yujini@kribb.re.kr (Y.-J.L.); jiyoun@kribb.re.kr (J.C.); 2KRIBB School of Bioscience, University of Science and Technology in Korea, Daejeon 34113, Korea; 3VORONOIBIO Inc., S 11th F, Thechnopark IT Center Songdo Kwahak-ro 32, Incheon 21984, Korea; namdoo@voronoibio.com

**Keywords:** antitumor, acacetin, STAT3 inhibitor, natural product, label-free methods

## Abstract

Signal transducer and activator of transcription 3 (STAT3) plays a critical role in the formation and growth of human cancer. Therefore, STAT3 is a therapeutic target for cancer drug discovery. Acacetin, a flavone present in various plants, inhibits constitutive and inducible STAT3 activation in STAT3-activated DU145 prostate cancer cells. Acacetin inhibits STAT3 activity by directly binding to STAT3, which we confirmed by a pull-down assay with a biotinylated compound and two level-free methods, namely, a drug affinity responsive target stability (DARTS) experiment and a cellular thermal shift assay (CETSA). Acacetin inhibits STAT3 phosphorylation at the tyrosine 705 residue and nuclear translocation in DU145 cells, which leads to the downregulation of STAT3 target genes. Acacetin then induces apoptosis in a time-dependent manner. Interestingly, acacetin induces the production of reactive oxygen species (ROS) that are not involved in the acacetin-induced inhibition of STAT3 activation because the suppressed p-STAT3 level is not rescued by treatment with GSH or NAC, which are general ROS inhibitors. We also found that acacetin inhibits tumor growth in xenografted nude mice. These results suggest that acacetin, as a STAT3 inhibitor, could be a possible drug candidate for targeting STAT3 for the treatment of cancer in humans.

## 1. Introduction

Cancer is a leading cause of death worldwide and its incidence is increasing continuously [1]. Signal transducer and activator of transcription 3 (STAT3) is a transcription factor that controls cell proliferation, differentiation, migration, invasion, apoptosis, and angiogenesis [2]. STAT3 is constitutively activated in many different types of cancers and tumor-infiltrating immune cells and has been associated with a poor prognosis [3,4]. Multiple studies suggest that targeting STAT3 in cancer suppresses cell growth, induces apoptosis, and enhances anticancer immune responses in vitro and in vivo, making it an attractive therapeutic target for cancer treatment [5,6]. Prostate cancer is the fourth most common cancer worldwide and the second most common cancer in men after lung cancer [1]. Elevated levels of IL-6 are observed in a large number of patients with solid tumors including prostate cancer, and IL6 stimulates the activation of STAT3 signaling pathway, which is often associated with poor patient outcomes [2]. Therefore, STAT3 is a potential target molecule for the development of therapeutic agents for prostate cancer.

Natural products (NPs) are isolated from live organisms, such as food and plants. NPs represent an unlimited source of drug discovery and more than 49% of antitumor drugs are derived, either directly or indirectly, from NPs [7,8,9]. Thus, there is increasing interest in the discovery of STAT3 inhibitors from NPs, and various types of NPs with STAT3 inhibitory activities have been reported [10]. Flavonoids are polyphenolic compounds and are widespread in a variety of fruits, vegetables, and seeds. They can be classified into flavanones, flavones, flavonols, and biflavones. Many flavonoids exhibit potent antitumor activity against several human cancer cell lines [11]. The flavone acacetin (5,7-dihydroxy-4′-methoxy flavone) has been reported to alleviate arthritis, obesity, asthma, neuroinflammation, and many types of cancer, including colorectal, breast, prostate, skin, and lung cancer, and leukemia [12,13]. Acacetin exerts an anticancer effect by regulating the activity of a wide variety of targets; its representative targets are PI-3K, VEGF, JNK, and STAT3 [14,15,16]. Although the target molecules of acacetin have been identified, the different modes of action associated with the binding target of acacetin are not yet completely understood.

One of the characteristics of NPs, including acacetin, is that they show anticancer activity by regulating various targets. Therefore, the validation of the target engagement of NPs is highly important in early drug discovery in order to improve the value of NPs. A number of experimental tools for NP target identification have been developed over the years, including various label-free methods [17].

In this study, we identified acacetin as a STAT3 inhibitor and verified that acacetin directly binds to STAT3 using various biochemical methods, including a pull-down assay, a drug affinity responsive target stability (DARTS) experiment, and a cellular thermal shift assay (CETSA), in DU145 prostate cancer cells with constitutively active STAT3 [18,19]. Acacetin was found to inhibit the tyrosine phosphorylation of STAT3 at tyrosine 705 in DU145 cells, which led to a blocking of the nuclear translocation of STAT3 and then the downregulation of the expression of STAT3-regulated genes. In addition to STAT3 inhibition, acacetin increased the levels of intercellular ROS. Therefore, acacetin showed antitumor activity in STAT3-activated DU145 cells via STAT3 inhibition and ROS generation.

## 2. Results and Discussion

### 2.1. Acacetin Suppresses the Proliferation of DU145 Cancer Cells by Reducing STAT3 Phosphorylation

To identify STAT3 inhibitors, we screened 650 natural products using a STAT3 luciferase reporter assay in HCT-116 colon cancer cells [20]. From the screening program, acacetin was identified as a STAT3 inhibitor (Figure 1A). Acacetin, an *O*-methylated flavone, was isolated from *Turnera diffusa* (damiana), *Betula pendula* (silver birch), *Asplenium normale* fern, and several other plants [21]. STAT3 is constitutively activated in DU145 cells and thus, the cells were used to validate the fact that STAT3 inhibitors regulate the phosphorylation of STAT3 and inhibit cell proliferation. To investigate the effect of acacetin on the proliferation of DU145 cells, acacetin was administrated at the indicated concentrations for 24, 48, or 72 h. Acacetin decreased the DU145 cell proliferation in a dose- and time-dependent manner with a GI_50_ value of 20 μM for 48 h treatment (Figure 1B). Acacetin significantly decreased the phosphorylation of STAT3 at Tyr-705 in a dose-dependent manner (Figure 1C). In a time-course treatment with acacetin, p-STAT3-Y705 was decreased up to 80% at 3 h and the total amount of STAT3 protein was maintained for up to 12 h (Figure 1D). To further evaluate the antitumor activity of acacetin, a colony formation assay was performed and acacetin was found to strongly inhibit the colony formation in a dose-dependent manner (Figure 1E). Acacetin (20 μM) was found to inhibit colony formation by more than 60% (Figure 1E). However, the phosphorylation of STAT3 by acacetin is reduced by about 60% with 50 μM treatment. Therefore, it is difficult to explain the anticancer effect of acacetin only by inhibiting the STAT3 activity, and it is thought that an anti-cancer effect may be shown by regulating multiple targets such as PI-3K, VEGF, and JNK.

As shown in Figure 1F, STAT3 was constitutively activated in DU145 cells but rarely activated in LNCaP (human prostate adenocarcinoma cells) and MCF10A (nontumorigenic epithelial cells) cells (Figure 1F). As STAT3 is activated by inflammatory cytokines such as IL-6, we assessed the amount of p-STAT3-Y705 present after the cells were treated with IL-6 (10 ng/mL) for 30 min [22]. STAT3 was strongly activated by the cytokine (Figure 1G); in particular, the treatment of LNCaP cells with IL-6 increased the amount of p-STAT3-Y705 by 22-fold. After the pretreatment of DU145 cells with acacetin, the IL-6-induced p-STAT3-Y705 in DU145, LNCaP, and MCF10A cells were abolished by more than 50%. Thus, acacetin inhibited constitutive STAT3 phosphorylation in DU-145 cells, as well as the IL-6-induced phosphorylation of STAT3 in LNCaP and MCF10A cells. HCC827 cells were treated with acacetin to measure the p-STAT3 levels, with results similar to those obtained by treatment with acacetin in DU145 being obtained (Appendix A). These results suggest that acacetin is a specific STAT3 inhibitor.

An important unmet need in the development of anticancer drugs is to affect cancer cells and not normal cells. Therefore, we selected the MCF10A and CCD-18Co (human colon fibroblasts) cell lines as normal cells and examined the expression levels of p-STAT3-Y705 in DU145, MCF10A, and CCD-18Co cells. The p-STAT3 levels in MCF10A and CCD-18Co cells were significantly lower than those in DU145 cells (Figure 1H). When DU145, MCF10A, and CCD-18Co cells were treated with 20 or 30 μM of acacetin, the cell viability of normal cells was higher than that of DU145 cells (Figure 1I). These results suggest that acacetin inhibits the growth of DU145 cells with high p-STAT3 levels more strongly than that of normal cells. Our results show that acacetin selectively reduces the phosphorylated STAT3 level, resulting in a strong inhibition of the colony formation of DU145 cells and an inhibition of cell growth dependent on the STAT3 activity.

### 2.2. Acacetin Reduces the Expression of STAT3 Target Genes and Induces Apoptosis

STAT3 is a transcription factor that regulates genes that are involved in proliferation and apoptosis [23]. Downstream target genes of STAT3, such as cyclin D1, Bcl-2, Mcl-1, Bcl-xL, and survivin, are involved in cell apoptosis and proliferation [24]. STAT3 is phosphorylated at the tyrosine-705 residue and forms homodimers, followed by translocation into the nucleus to regulate the expression of target genes [2,3]. We found that acacetin suppressed the nuclear translocation of STAT3 after 24 h of treatment in DU145 cells (Figure 2A). After 24 h of treatment with acacetin, acacetin inhibited the expression of STAT3 target proteins, such as cyclin D1, Bcl-2, Bcl-xL, Mcl-1, and survivin (Figure 2B); however, Bax, an apoptotic marker protein, was increased in acacetin-treated cells. Moreover, we evaluated the effect of acacetin on the cell cycle progression in DU145 cells using a FACSCalibur flow cytometer. As shown in Figure 2C, when acacetin was added to DU145 cells, the proportion of sub-G1 phase cells was increased, resulting in an increase in cleaved PARP. Then, we used an Annexin V-FITC assay to quantitatively evaluate the percentage of apoptotic cells and found that treatment with acacetin for 24 or 48 h increased the percentage of apoptotic cells by 2.4% and 4%, respectively (Figure 2D). Our results suggest that acacetin induces cell apoptosis by downregulating STAT3 target genes, which are involved in pro- and antiapoptotic proteins. Recent studies have indicated that NPs can be used as anticancer therapies with relatively low toxicity [9]. Although there are many problems in the clinical application of NPs, such as their low aqueous solubility, low metabolic stability, and low bioavailability, these problems are expected to be improved quickly with the development of nanotechnology and drug delivery tools [25]. Therefore, our results suggest that acacetin could be a potent anticancer agent targeting STAT3-activated cancer cells.

### 2.3. Acacetin Induces ROS Generation

Many flavanones induce the apoptosis of cancer cells via activation of the mitochondrial apoptotic pathway by increasing ROS production [26]. It was reported that a STAT3 inhibitor generates ROS and leads to the inactivation of STAT3 signaling and the induction of apoptosis in human colon cancer HCT116 cells [27]. 2-Hydroxycinnamaldhyde (HCA), a phenolic NP, induces apoptosis via ROS generation in human cancer cells, which is inhibited by ROS scavenger treatment [28]. We also found that HCA inhibits STAT3 activity by generating ROS in DU145 cells [29]. We measured the ROS generation induced by acacetin treatment using a FACSCalibur flow cytometer and fluorescence confocal microscope. The ROS induced by acacetin increased at 1 h and persevered for up to 6 h (Figure 3A,B). To determine whether acacetin-induced ROS generation is related to p-STAT3, DU145 cells were treated with ROS scavengers such as *N*-acetyl-l-cysteine (NAC) and glutathione (GSH) for 1 h, then treated with acacetin. However, the acacetin-induced decrease in p-STAT3 was not rescued by the pretreatment of the DU145 cells with NAC or GSH (Figure 3C). Furthermore, STAT3 inhibitors and compounds that raise ROS levels increase p-ERK levels, and NAC or GSH treatment tends to a decrease in the elevated p-ERK levels [29]. However, in the case of acacetin, it is not known well, but this shows that the p-ERK/ERK level increases slightly when acacetin is administered and increases even more when NAC is administered with it. More research is needed in order to understand this mechanism. Therefore, these results suggest that acacetin-mediated ROS generation is not affected by p-STAT3-Y705.

### 2.4. Acacetin Does Not Inhibit Upstream Kinases of STAT3

Several tyrosine kinases, including EGFR, JAK2, and mTOR, have been reported to phosphorylate STAT3, and STAT3 has been reported to be activated by tyrosine kinases of the JAK family [30]. Therefore, we tested whether acacetin inhibits these kinases in vitro. As shown in Figure 4A, acacetin did not inhibit the kinase activity of EGFR and only partially inhibited the activity of JAK2 and mTOR at a dose of 5 μM or 10 μM in vitro. Therefore, we assessed whether acacetin inhibited JAK2, DU145 cells were treated with acacetin at 50 μM for up to 3 h and the level of p-JAK2 was measured using p-JAK2-specific antibodies. Unexpectedly, it was found that the level of p-JAK2 increased rather than decreased and that acacetin did not inhibit JAK2 expression; this mechanism needs to be elucidated through further study (Figure 4B). These results suggest that the upstream tyrosine kinase families regulating STAT3 phosphorylation may not be the molecular targets of acacetin. Therefore, further studies are required to elucidate the precise mechanism of the STAT3 inhibitory activity of acacetin in DU145 cells.

### 2.5. Acacetin Inhibits STAT3 Activity by Directly Binding to STAT3

To determine how acacetin regulates STAT3 activity, several experiments were performed based on the results of previous studies. Protein tyrosine phosphatases are well-known negative regulators of STAT3 activation [31]. It was reported that NPs inhibit the STAT3 phosphorylation by regulating the expression or activation of protein tyrosine phosphatases [32,33]. Therefore, we investigated whether the acacetin-mediated suppression of STAT3 phosphorylation could be due to the activity of a protein tyrosine phosphatase (PTP). The treatment of DU145 cells with sodium pervanadate, a broad-acting tyrosine phosphatase inhibitor, did not have an effect on the acacetin-induced inhibition of p-STAT3-Y705 (unpublished data), indicating that PTPs are not involved in acacetin-induced STAT3 dephosphorylation.

There have been reports that some STAT3 inhibitors bind directly to STAT3 and inhibit STAT3 phosphorylation within 1 h [20,29,34]. Acacetin also inhibited STAT3 phosphorylation within 1 h (Figure 1D). To assess the direct binding of acacetin to STAT3, we synthesized biotinyl–apigenin, in which only methyl groups were removed from acacetin (Figure 5A). To test whether biotin-apigenin decreases STAT3 phosphorylation in a manner similar to acacetin, DU145 cells were treated for 6 h. Biotin-apigenin was found to decrease the level of p-STAT3-Y706 by 90% at 50 μM (Figure 5B), indicating that it maintained its biological activity and could be used for pull-down assays. Lysates of DU145 cells were incubated with acacetin for 1 h and then incubated with biotin-apigenin at room temperature for 1 h. The bound proteins were precipitated with NeutrAvidin-agarose resin, resolved via SDS-PAGE and immunoblotted with antibodies against STAT3 or GAPDH (Figure 5C). Biotin-apigenin binds to STAT3, and its binding was reduced competitively by acacetin.

We also analyzed the binding site of acacetin in STAT3 using a computational docking study. As shown in Figure 5D, acacetin interacts with SH-2 domain of STAT3 via three hydrogen bonds (green dotted line) and a cation–π interaction (red dotted line).

Additionally, we confirmed the binding of acacetin to STAT3 using label-free methods, such as DARTS and CETSA. DARTS is based on the principle that a small-molecule drug will stabilize the structure of a target protein and cause protease resistance [35]. Acacetin weakly enhanced the protease resistance to pronase (Figure 6A) and STAT3 accumulation was increased in a dose-dependent manner with 0.005% pronase (Figure 6B). CETSA is based on the idea that the stability of proteins is affected by the binding of small molecules to proteins, and protein thermal stability assays have been used for many years in drug discovery programs, including for STAT3 inhibitors [36]. After the treatment of DU145 cell lysates with acacetin, the thermal stability of STAT3 decreased, with an increased melting temperature (Figure 6C). At 50 °C, acacetin decreased the thermal stability of STAT3 in a dose-dependent manner; however, acacetin did not change the STAT3 level at 37 °C (Figure 6D). Collectively, acacetin specifically binds to STAT3 in DU145 cells, showing the increased pronase resistance and decreased thermal stability of STAT3. These data indicate that acacetin binds directly to STAT3 and then selectively and specifically inhibits the proliferation of human cells that exhibit constitutively active STAT3.

### 2.6. Acacetin Suppresses Tumor Growth in a Mouse DU145 Cell Xenograft Model

To investigate the effect of acacetin in vivo, we used a mouse DU145 cell xenograft model. The flank of each nude mouse was injected with 9 × 10^6^ DU145 cells. Beginning one day after tumor challenge, vehicle or acacetin (50 mg/kg) was intraperitoneally injected 5 days per week for 30 days. On day 30, the mice were sacrificed, and the tumors were removed and weighed. Acacetin significantly suppressed tumor growth, with a 67.3% decrease in the tumor volume and a 67.9% decrease in the tumor weight of mice xenografted with DU145 cells compared with vehicle-treated mice (Figure 7A,B). Some differences in body weight were observed between the vehicle control and acacetin-treated mice (Figure 7D). Zhou et al. reported on the possibility of toxicity through cytochrome P450 inhibition when acacetin is administered intraperitoneally at a dose of 50 mg/kg in rats, but there has been no study on the exact toxicity, meaning that more research is needed [37]. These results indicate that acacetin inhibits the in vitro cell proliferation and tumor growth of STAT3-activated DU145 prostate cancer cells, which suggests the potential of acacetin as an anticancer agent for STAT3-activated tumor cells.

## 3. Materials and Methods

### 3.1. Chemicals and Reagents

Acacetin was purchased from Ambeed (Arlington Hts, IL, USA). N-biotinylcaproic acid was obtained from Toronto Research Chemicals (Toronto, Canada). Other chemicals were purchased from Sigma-Aldrich Chemical Co. Ltd. (St. Louis, MO, USA) and Tokyo Chemical Industry (Tokyo, Japan). Fetal bovine serum (FBS), Dulbecco’s modified Eagle’s medium (DMEM), antibiotics (10,000 U/mL penicillin, 10,000 μg/mL streptomycin), and RPMI 1640 medium were purchased from Gibco (Gland Island, NY, USA). Propidium iodide (PI) was purchased from Sigma-Aldrich Chemical Co., Ltd. (St. Louis, MO, USA). Phosphate-buffered saline (PBS) and RIPA lysis buffer were purchased from LPS solution (Daejeon, Korea).

The antibodies used were purchased from Cell Signaling (Danvers, MA, USA) and targeted p-STAT3 (Y705), STAT3, BCL-2, Bax, Bcl-xL, MCl-1, Survivin, PARP, JAK2, p-p38, and p38. Additional antibodies against cyclin D1, cyclin A, and p-JAK2 and GAPDH; goat-anti-rabbit IgG-FITC secondary antibody; and DAPI were purchased from Santa Cruz (Dallas, TX, USA). The secondary antibodies used were horseradish peroxidase-conjugated goat anti-rabbit or anti-mouse IgG (Jackson ImmunoResearch Laboratories, West Grove, PA, USA). Annexin V-FITC was purchased from BD Pharmingen (San Diego, California, USA). PVDF (polyvinylidene fluoride) membranes (0.45 μm) were purchased from EMD Millipore (Billerica, MA). NeutrAvidin beads were purchased from Thermo Fisher Scientific (Waltham, MA, USA, 29202). Protease inhibitor cocktail solution (100×) was purchased from Gen-Depot (Barker, TX, USA). Pronase was purchased from Roche Diagnostics (Rotkreuz, Switzerland).

### 3.2. Cell Lines and Culture Conditions

All the cell lines used in this study were originally obtained from ATCC (Manassas, VA, USA). DU145 and LNCaP (human prostate cancer) were maintained in RPMI 1640 medium (Gibco, Gaithersburg, MD, USA). MCF-10A (human mammary epithelial) and CCD-18Co (human colon fibroblast) were maintained in DMEM-F12 (Gibco). All culture media were supplemented with 10% heat-inactivated FBS (Gibco), 100 U/mL of penicillin, and 0.1 mg/mL of streptomycin (Sigma-Aldrich, St. Louis, MO, USA). Cell cultures were maintained in a 37 °C incubator under a humidified atmosphere with 5% CO_2_.

### 3.3. Cell Proliferation Assay

Cells were seeded in 24-well plates in medium containing 10% FBS. After 24 h, the wells were replenished with fresh complete medium containing either a test compound or 0.1% DMSO. After incubation for 24–48 h, the medium was collected, and the cells were trypsinized. Cells were stained with trypan blue and then the cell number was determined using a hemocytometer. Each sample was evaluated in triplicate.

### 3.4. Colony Formation Assay

DU145 cells were seeded on 6-well plates, and the cells were treated with a compound after 4–6 h. After incubation for 2–3 days, the wells were replenished with fresh complete medium. After 5–7 days, the cells were fixed with 5 mg/mL of crystal violet containing 20% methanol at 25 °C for 30 min. Stained cells were washed gently with 1X PBS, and air dried at room temperature. The number of colonies present was quantified using a microscope (Nikon Eclipse TE300; Nikon, Tokyo, Japan).

### 3.5. Western Blot Analysis

Cell lysates were prepared in RIPA lysis buffer containing a protease inhibitor cocktail (Roche Diagnostics). Proteins (20–50 μg) were resolved in 8–15% SDS-PAGE gels and transferred to PVDF membranes (EMD Millipore, Billerica, MA, USA). The membranes were blocked with 5% nonfat dry milk in TBST and incubated with primary and secondary antibodies according to the manufacturer’s protocols. The secondary antibodies used were horseradish peroxidase-conjugated goat anti-rabbit or anti-mouse IgG (Jackson ImmunoResearch Laboratories, West Grove, PA, USA). The membranes were washed three times with TBST, then incubated with Luminata Forte Western HRP substrate (EMD Millipore). The signal was detected using a LAS 4000 mini system (GE Healthcare Life Sciences, Pittsburgh, PA, USA). The densitometric analysis of the bands was performed using the MultiGauge program (Fuji Photo Film Co, Ltd., Tokyo, Japan), and the results were normalized to the corresponding GAPDH levels.

### 3.6. Fluorescence-Activated Cell Sorting (FACS) Analyses

Cells were trypsinized at specific times after treatment with specific compounds and were collected by centrifugation at 1080 rpm for 5 min at room temperature. The supernatant was discarded, and the precipitated cells were washed twice by repeated suspension in PBS. Precipitated cells were carefully suspended in 500 μL of PBS buffer and fixed with 4 mL of ice-cold 70% ethanol overnight. Fixed cells were washed twice with PBS. The collected cells were resuspended in PBS and treated with 100 μg/mL RNase A at 37 °C for 30 min. Propidium iodide was then added at a final concentration of 50 μg/mL for DNA staining. Fixed cells were analyzed on a FACSCalibur flow cytometer until 20,000 cells had been counted (BD Biosciences, San Jose, CA, USA). The distribution of the cells across the cell cycle was analyzed using the WinMDI 2.9 software (BD Bioscience, San Diego, CA, USA).

### 3.7. Apoptosis Assay Using Annexin V Staining

Assays were carried out using the FITC-annexin V apoptosis detection kit (BD Biosciences). DU-145 cells were trypsinized and collected by centrifugation at 1080 rpm for 5 min at room temperature. Precipitated cells were washed with PBS and double-stained with annexin V-FITC and propidium iodide. Apoptotic cells were analyzed with a FACSVerse flow cytometer (BD Biosciences).

### 3.8. ROS Measurement

A FACSCalibur flow cytometer (BD Biosciences) was used to assess the amounts of ROS. For green fluorescence, the excitation wavelength was 488 nm, and the observation wavelength was 530 nm. The relative change in fluorescence was analyzed with the WinMDI software. For the measurement of intracellular ROS levels, detached cells were incubated with 5 μM CM-H_2_DCFDA for 30 min at 37 °C.

ROS were also measured via the DCFDA fluorescence staining method. DU145 cells (1 × 10^5^) were plated into 35 mm μ-dishes and treated with DMSO or acacetin (50 μM) for the indicated times. Treatment with 500 μM H_2_O_2_ for 30 min was used as a positive control. The treated cells were incubated with 10 μM of DCFDA for 50 min. The signal was detected via laser scanning confocal microscopy with excitation/emission at 485/535 nm.

### 3.9. Synthesis of Biotin-Apigenin

A mixture of N-biotinylcaproic acid (100 mg, 1.0 equiv.), 1-(3-dimethylaminopropyl)-3-ethylcarbodiimide hydrochloride (EDC, 100 mg, 1.7 equiv.), and *N*-dimethylaminopyridine (DMAP, 20 mg) was dissolved in 50 mL of dimethylformamide, to which 100 mg of apigenin (1.3 equiv.) was added. The reaction mixture was stirred for 1 h at room temperature under nitrogen to completely dissolve the components and then stirred overnight under nitrogen at room temperature. Next, the reaction solution was diluted with methylene chloride and water. The organic layer was dried over anhydrous MgSO4 and filtered through filter paper. The filtrate was concentrated in vacuo and purified via silica gel flash column chromatography to provide the desired product apigenin-biotin (49 mg, yield: 28%). HR ESI *m*/*z*: [M + H]^−^ calcd for C_31_H_36_N_3_O_8_S, 610.2223; found, 610.2215. ^1^H NMR (800 MHz, DMSO-d_6_) δ 12.77 (m, 1H), 10.83 (m, 1H), 8.10 (m, 2H), 7.73 (s, 1H), 7.30 (m, 2H), 6.93 (m, 1H), 6.49 (m, 1H), 6.40 (s, 1H), 6.33 (s, 1H), 6.19 (m, 1H), 4.29 (m, 1H), 4.12 (m, 1H), 3.06 (m, 3H), 2.80 (m, 1H), 2.59 (m, 3H), 2.05 (m, 2H), 1.3~1.7 (m, H12). ^13^C NMR (DMSO-d_6_) δ 181.72, 171.80, 171.26, 164.30, 162.64, 162.34, 161.44, 157.35, 153.11, 128.17, 127.79, 122.46, 105.10, 103.87, 98.96, 93.97, 60.97, 59.15, 55.40, 38.11, 35.19, 33.40, 30.70, 28.77, 28.12, 27.94, 25.75, 25.27, 23.92.

### 3.10. Pull-Down Assay

DU145 cells were washed with PBS and homogenized in binding buffer (10 mmol/L Tris-HCl, pH = 7.4, 50 mmol/L KCl, 5 mmol/L MgCl_2_, 1 mmol/L EDTA, and 0.1 mmol/L Na_3_VO_4_) with a 26-gauge syringe. The cell lysate was centrifuged, and the supernatant was collected. The cell lysate was precleared by incubation with NeutrAvidin beads (Thermo Fisher Scientific, 29202) for 1 h at 4 °C. The cleared lysate was incubated with biotin-conjugated apigenin (biotin-apigenin) for 1 h at 4 °C. Proteins bound to biotin–apigenin were precipitated with NeutrAvidin beads. After 3 washes in washing buffer (50 mmol/L HEPES, pH 7.5, 50 mmol/L NaCl, 1 mmol/L EDTA, 1 mmol/L EGTA, 0.1% Tween-20, 10% (*v*/*v*) glycerol, 1 mmol/L NaF, 0.1 mmol/L Na_3_VO_4,_ and 1× protease inhibitor cocktail (Roche Diagnostics), the beads were eluted with 1× sample buffer. The samples were boiled for 10 min and separated for Coomassie blue staining or immunoblotting.

### 3.11. Drug Affinity Responsive Target Stability (DARTS)

The DARTS experiment was conducted as previously described with some modifications [18]. Cells were washed with ice-cold PBS and treated with ice-cold M-PER lysis buffer (Thermo Fisher Scientific Inc., Rockford, IL, USA) supplemented with a protease inhibitor cocktail, 1 mM Na_3_VO_4_ and 1 mM NaF. The protein lysates were mixed with 10× TNC buffer (500 mM Tris-HCl at pH = 8.0, 500 mM NaCl, and 100 mM CaCl_2_). The lysates in 1× TNC buffer were incubated with DMSO or acacetin for 1 h at room temperature. Following incubation, each sample was proteolyzed in various concentrations of pronase (Roche Diagnostics, 10,165,921,001) for 10 min at room temperature. After 10 min, 2 μL of ice-cold 20 × protease inhibitor cocktail was added to stop proteolysis, and the samples were immediately placed on ice. Digestion was further stopped by the addition of 5× sample loading dye and boiling at 95 °C for 10 min. An equal portion of each sample was then loaded onto SDS-PAGE gels for Western blotting.

### 3.12. Cellular Thermal Shift Assay (CETSA)

A CETSA was conducted using cell lysates as previously described [28]. For CETSAs with cell lysates, DU145 cells were lysed with lysis buffer (50 mM Tris-HCl, pH = 7.5, 100 mM NaCl, 0.2% NP-49, 5% glycerol, 1.5 mM MgCl_2_, 25 mM NaF, 1 mM Na_3_VO_4_, and 1 × protease inhibitor cocktail). After centrifugation, the lysates were incubated with DMSO or acacetin for 1 h at room temperature. The lysates were aliquoted into 0.2 mL PCR tubes and heated for 5 min at the indicated temperature in a PCR machine (Applied Biosystems). The precipitated proteins were separated from the soluble fraction by centrifugation and equal portions of the supernatants were loaded onto SDS-PAGE gels for Western blotting.

### 3.13. Kinase Assay

The kinase assay was carried out by Merck Millipore (Burlington, MA, USA). Protein kinases were tested in a radiometric assay format, and the raw data were measured by scintillation counting (in cpm). For kinase dilution and reaction, the following buffer composition was used: 20 mM MOPS, 1 mM EDTA, 0.01% Brij-35, 5% glycerol, 0.1% mercaptoethanol, and 1 mg/mL BSA.

### 3.14. Immunocytochemistry

DU145 cells (1.0 × 10^5^ cells) were plated into 35 mm high μ dishes (ibidi GmbH, Am Klopferspitz, Germany). The cells were washed once with PBS and treated with DMSO or acacetin (50 μM) for 24 h. After washing with PBS twice, the attached cells were fixed with 4% paraformaldehyde in PBS for 10 min at room temperature. The fixed cells were permeabilized with 0.2% Triton X-100 for 10 min and blocked with 1.0% BSA in PBS for 1 h. The cells were incubated with an anti-STAT3 antibody followed by goat-anti-rabbit IgG-FITC secondary antibody. The nuclei were counterstained with 2 μg/mL of DAPI in PBS for 2 min. All images were acquired on a laser scanning confocal microscope (LSM 510 META; Carl Zeiss, St. Cloud, MN, USA) and analyzed with the LSM Version 3.2 software (Carl Zeiss).

### 3.15. In Vivo Xenograft Assay

Animal experiments were performed in accordance with a protocol approved by the Korea Research Institute of Bioscience and Biotechnology (KRIBB) Animal Experimentation Ethics Committee. DU145 cells (9 × 10^6^ cells/mouse) were subcutaneously injected into the right flank of nude mice (6-week-old female BALB/c mice; *BALB*/*c*-Foxn1 nu/CrljOri). Tumor growth was measured with calipers using the formula volume = 1/2 (length (mm) × width (mm) × height (mm)). Vehicle and acacetin (50 mg/kg) were intraperitoneally injected 5 days per week for 30 days starting on day 1 (the day after cell injection; six mice per group). On day 30, the mice were sacrificed and the tumors were removed and weighed (mg). Tumor measurements were performed every 3 days with a caliper.

### 3.16. Computational Molecular Docking

For the prediction of the docking model of acacetin with STAT3 SH2 domain, we used the crystal structure of STAT3 (PDB code; 6NJS). The 3D structure of acacetin was built using a Maestro build panel. The acacetin was minimized using the MacroModel tool of Maestro in the Schrödinger Package (http://www.schrodinger.com). The STAT3 crystal structure was minimized using the Protein Preparation Wizard of Maestro with the OPLS3e force field. Molecular docking was performed with the Glide tool using a receptor grid localized within the STAT3 SH2 domain. Molecular graphics for the docking model of the acacetin with STAT3 SH2 domain were generated using the PyMol package (http://www.pymol.org).

### 3.17. Statistical Analysis

All experiments were performed at least twice, and multiple samples represented biological (not technological) replicates. All animal experiments were performed using randomly assigned groups without investigator blinding. No statistical methods were used to predetermine the sample size, and no data were excluded. Statistical analyses were performed using Excel. Statistical significance was tested using two-tailed *t*-tests. Values of *p* < 0.05, *p* < 0.01, and *p* < 0.001 are denoted by *, **, and ***, respectively.

## 4. Conclusions

Due to the advantages of NP-based drugs with low toxicity, several natural STAT3 inhibitors have been developed as potential anticancer therapeutics including curcumin, napabucasin, cryptotanshinone, butein, capsaicin, celastrol, diosgenin, thymoquinone, resveratrol, piperlongumine, and vitamin E [10]. In this study, we found that acacetin is a natural STAT3 inhibitor, and its functions as a STAT3 inhibitor were shown by direct binding to STAT3 proteins. Acacetin inhibits the translocation of STAT3 into the nucleus, resulting in selectively suppressed STAT3-activated cell proliferation and the induced apoptosis of DU145 prostate cancer cells. In mice xenografted with DU145 cells, acacetin inhibited the tumor volume and the tumor weight. Therefore, our results suggest that acacetin could be a potent anticancer agent for targeting STAT3-activated tumor cells.

## Figures and Tables

**Figure 1 molecules-26-06204-f001:**
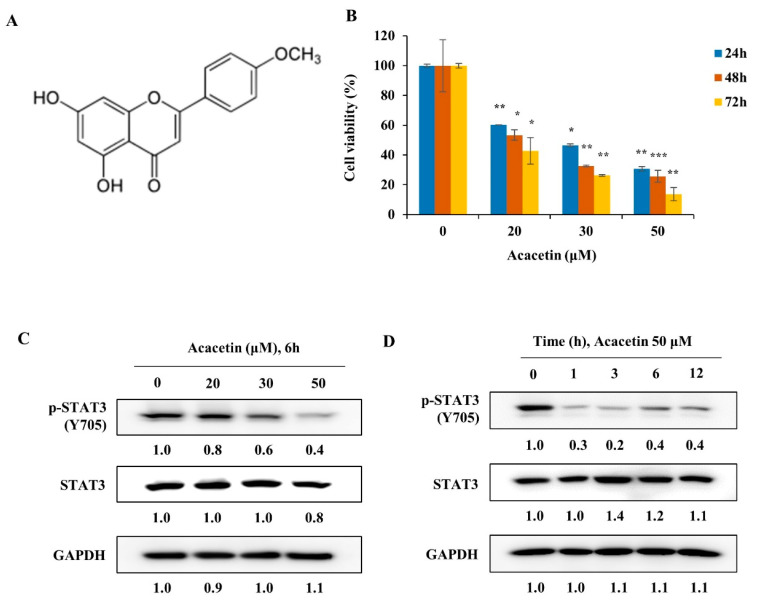
Acacetin inhibits p-STAT3-Y705 in a dose- and time-dependent manner in prostate cancer cells. (**A**) The structure of acacetin (Aca). (**B**) Acacetin induced the growth inhibition of DU145 cells. Cells were plated in triplicate and treated with the indicated concentrations of acacetin for 24, 48, or 72 h. Cell proliferation was measured using manual cell counts (*n* = 3). (**C**) DU145 cells were treated with acacetin at the indicated concentration for 6 h; whole lysates were obtained and analyzed by Western blotting with p-STAT3 (Y705) and STAT3 antibodies. (**D**) DU145 cells were treated with acacetin (50 μM) for the indicated time and analyzed by Western blotting with p-STAT3 (Y705) and STAT3 antibodies. (**E**) A colony formation assay was performed in DU145 cells exposed to acacetin at different concentrations. Scale bars, 1 cm. (**F**) The protein expression levels of p-STAT3 (Y705) and STAT3 in DU145, LNCaP, and MCF10A cells (*n* = 2). (**G**) DU145, LNCaP, and MCF 10A cells were stimulated with IL-6 (10 ng/mL) for 30 min in the presence or absence of acacetin. The expression levels of p-STAT3 (Y705) and STAT3 were analyzed by Western blotting (*n* = 2). (**H**) Expression levels were analyzed by Western blotting with p-STAT3 (Y705), STAT3, and GAPDH antibodies. (**I**) DU145, MCF10A, or CCD-18Co cells were treated with 20 or 30 μM for 24 h. Cell proliferation measured using manual cell counts (*n* = 3). The band intensity of each protein was quantified using the Multi Gauge program. The data represent the means ± s.d.; comparisons were performed with *t*-tests; * *p* < 0.05, ** *p* < 0.01, *** *p* < 0.001.

**Figure 2 molecules-26-06204-f002:**
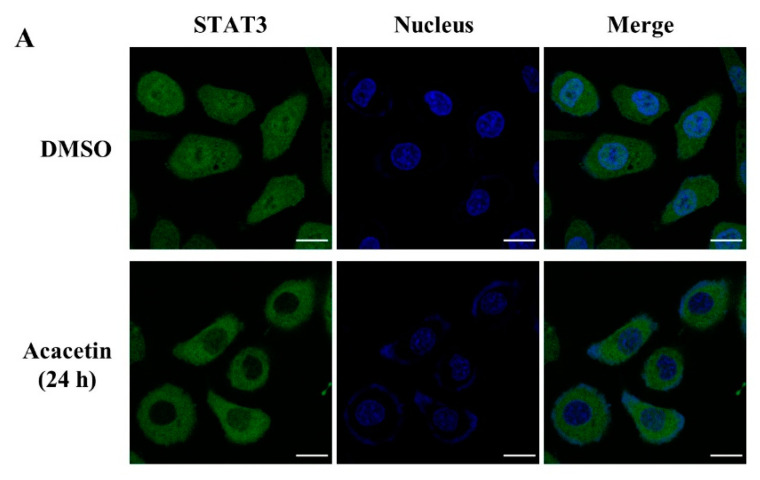
Acacetin inhibits the translocation of STAT3 into the nucleus and downregulates the expression of STAT3 target genes that induce apoptosis. (**A**) DU145 cells were treated with acacetin (50 μM) for 24 h. The nuclear localization of STAT3 was analyzed by immunocytochemistry using STAT3 antibody (green). Scale bars, 20 μm. (**B**) DU145 cells were treated with acacetin (50 μM) for 24 h, and the expression level of STAT3 target genes was analyzed by Western blotting. The band intensity of each protein was quantified using the MultiGauge program. (**C**) DU145 cells were treated with acacetin (50 μM) for the indicated times. After treatment, the cell cycle distribution was analyzed using a FACSCalibur flow cytometer. The ratios of cells in each phase were analyzed using the WinMDI 2.9 analyzer. (**D**) DU145 cells were treated with acacetin for 24 or 48 h and dual-stained with annexin V-FITC and propidium iodide. Apoptotic cell death was detected by flow cytometry. The data represent the means ± s.d.; comparisons were performed with *t*-tests; * *p* < 0.05.

**Figure 3 molecules-26-06204-f003:**
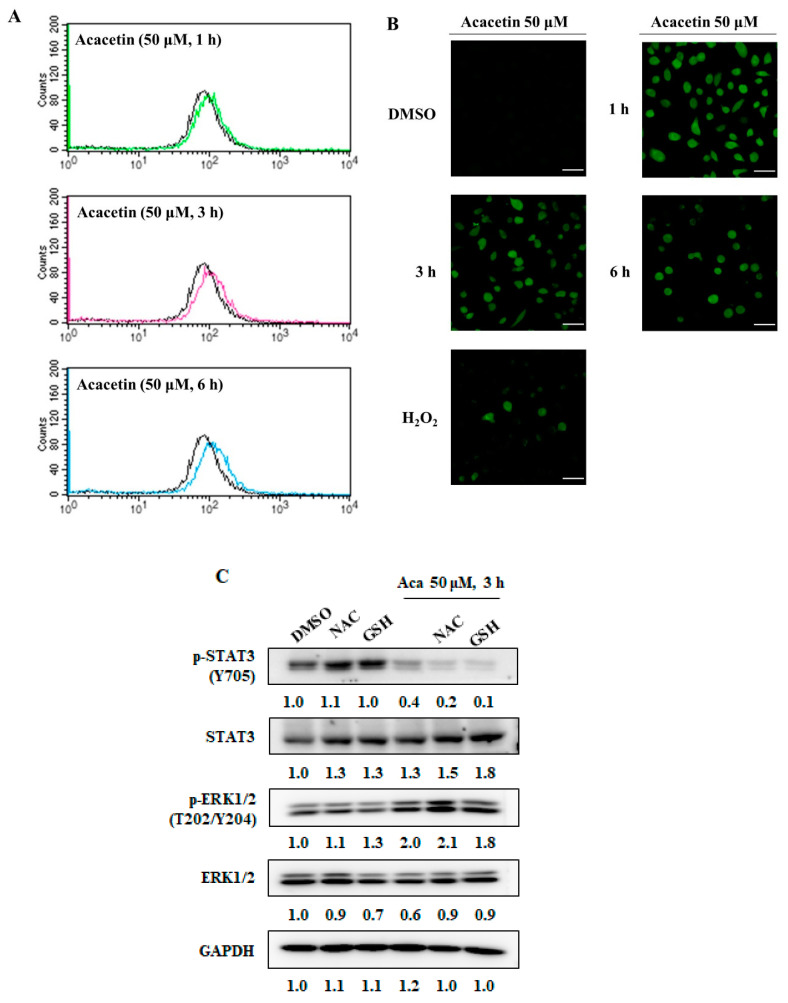
Acacetin promotes ROS generation. (**A**,**B**) DU145 cells were treated with acacetin (50 μM) for 1, 3, or 6 h. After treatment, the intracellular ROS levels were measured with DCF-DA (10 μM) using a FACSCalibur flow cytometer (**A**) and laser scanning confocal microscope. Scale bars, 50 μm (**B**). (**C**) After pretreatment with NAC (5 mM) or GSH (5 mM) for 1 h, DU145 cells were treated with acacetin (50 μM) for 3 h in the presence or absence of NAC or GSH. The expression levels of p-STAT3 (Y705) and STAT3 were analyzed by Western blotting. The band intensity of each protein was quantified using the MultiGauge program.

**Figure 4 molecules-26-06204-f004:**
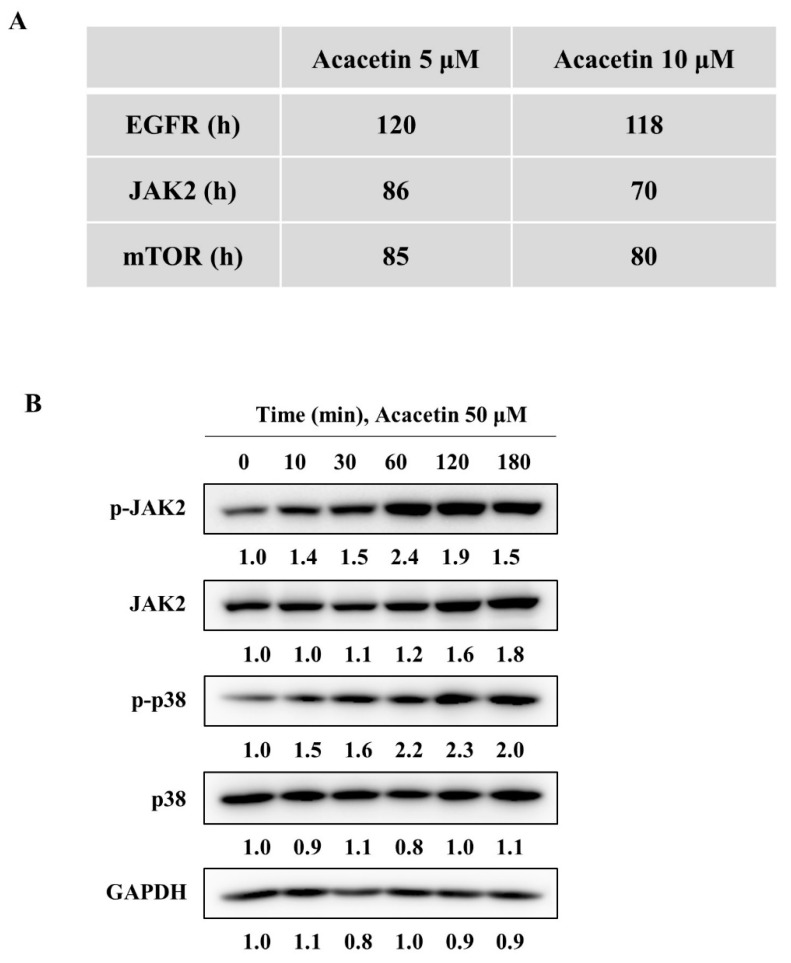
Acacetin does not inhibit STAT3 kinases. (**A**) In vitro kinase assay of known STAT3 kinases in the presence of 5 μM or 10 μM acacetin. (**B**) DU145 cells were treated with acacetin (50 μM) for the indicated time and analyzed by Western blotting with p-JAK2, JAK2, p-p38, and p38 antibodies. The band intensity of each protein was quantified using the MultiGauge program.

**Figure 5 molecules-26-06204-f005:**
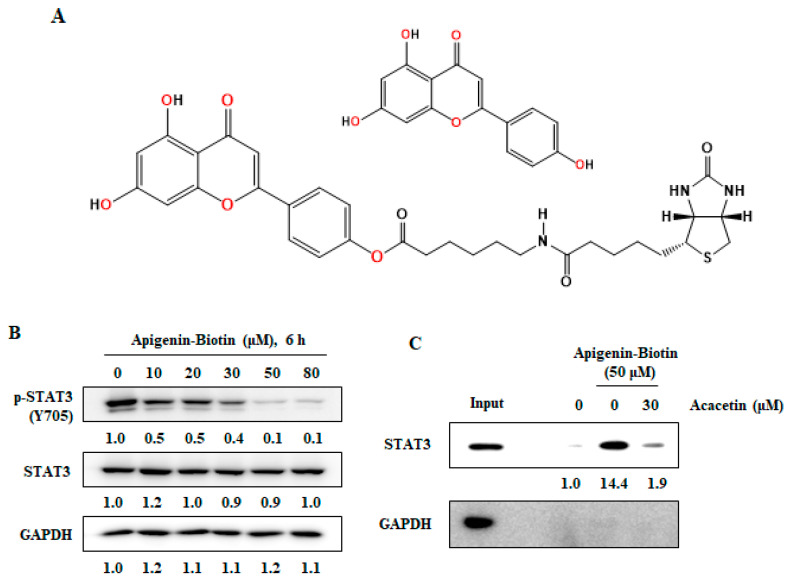
Acacetin binds to STAT3 and inhibits STAT3 activity. (**A**) Structure of apigenin and biotinyl–apigenin. (**B**) Biotinyl–apigenin induced the dephosphorylation of STAT3 in DU145 cells. DU145 cells were treated with the indicated concentration of biotinyl–apigenin for 6 h. Whole-cell lysates were analyzed using Western blotting. (**C**) DU145 lysates were incubated with DMSO or biotinyl–apigenin and then competed with acacetin at the indicated concentration. Binding protein was captured by NeutrAvidin agarose resin and eluted by boiling in SDS-PAGE sample buffer. The eluted protein was analyzed using Western blotting. (**D**) Structural model for acacetin in complex with STAT3. Acacetin is colored yellow, and hydrogen bonds between acacetin and STAT3 are shown as green dotted lines, and a cation–π interaction is shown as a red dotted line.

**Figure 6 molecules-26-06204-f006:**
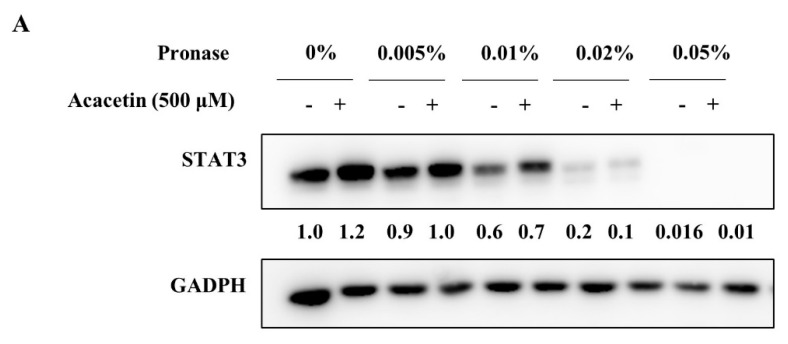
Acacetin directly binds to STAT3 in DU145 cells. (**A**) DU145 cell lysates were incubated with acacetin (500 μM) for 1 h at room temperature and then proteolyzed with increasing concentrations of pronase for 10 min (*n* = 2). (**B**) DU145 cell lysates were incubated with increasing concentrations of acacetin for 1 h at room temperature and then were proteolyzed with pronase for 10 min (*n* = 2). The pronase-resistant proteins in the presence or absence of acacetin were analyzed by Western blotting. (**C**,**D**) CETSA was performed with cell lysates of DU145 cells that were treated with DMSO or acacetin (500 μM) for 1 h with an increasing melting temperature (42–64 °C, interval temperature: 2 °C) (**C**) or drug concentration (**D**) (*n* = 2). The band intensity of each protein was quantified using the MultiGauge program. The data represent the means ± s.d.; comparisons were performed with *t*-tests; * *p* < 0.05, ** *p* < 0.01.

**Figure 7 molecules-26-06204-f007:**
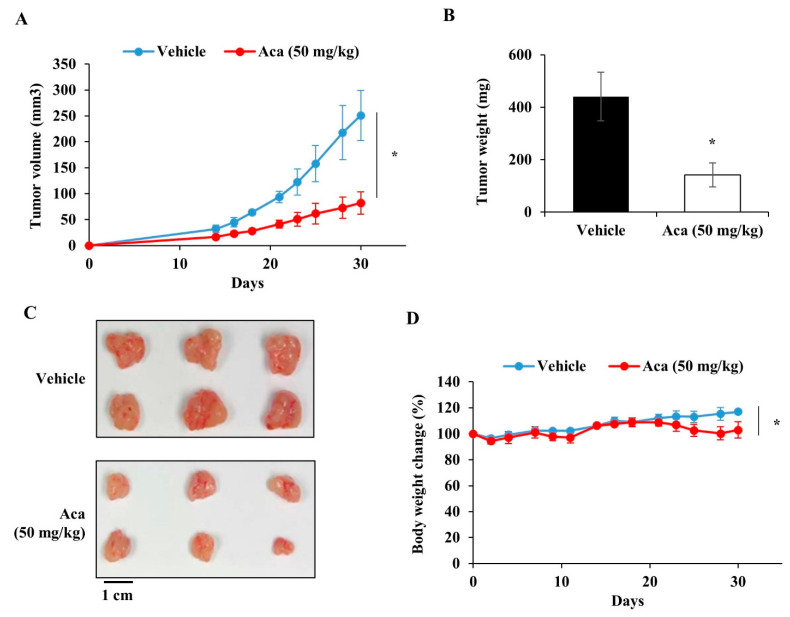
Acacetin inhibits the growth of DU145 human prostate cancer cells in a nude mouse xenograft model. (**A**) Quantification of the tumor volumes in each mouse (*n* = 6 per group) on day 30. (**B**) Quantification of the tumor weight of tumor from each mouse (*n* = 6 per group) on day 30. (**C**) Representative image of tumor tissues. (**D**) Evaluation of the body weight for each mouse (*n* = 6 per group). The data represent the means ± s.d.; comparisons were performed with *t*-tests; * *p* < 0.05.

## Data Availability

Not applicable.

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
