# Peer review of "Acacetin Inhibits the Growth of STAT3-Activated DU145 Prostate Cancer Cells by Directly Binding to Signal Transducer and Activator of Transcription 3 (STAT3)"

_molecules, 2021, doi:10.3390/molecules26206204_

Round 1

Reviewer 1 Report

Authors aim to demonstrate that acacetin by inhibiting STAT-3 posphorylation regulates growth and migration.

There are several concerns:

It is already reported in literature that acacetin inhibits stat-3 posphorylation.

Introduction could be improved focusing on PC Cancer and nutraceuticals.

See for example https://doi.org/10.3390/nu12092648

Even if the manuscript corroborate the acacetin control of STAT-3 P, tue experiments reported do not support their conclusions. Acacetin modulates growth, however it is not demonstrated the pivotal role of STAT-3 P. As an example, acacetin at 20 uM modulates growth but have no effect on STAT-3 P (see FIG.1C).

Functional studies, as growth assay and colony and etc should be performed with STAT-3 mutants that are not phosporylated and, if exists with inhibitors of stat-3 phosphorylation.

What is the level of stat-3 P in other prostate cell lines?

Are there any correlations with AR and/or ER?

In method section IF is lacking. What is the localization of STAT-3P?

Results in FIG.2B seems not statisticaL significant.

Author Response

Attached a file for autor's notes to reviewer

Reviewer 2 Report

Yun et al. explore that acacetin reduces the phosphorylated STAT3 level, resulting in strongly inhibiting the colony formation of DU145 cells and inhibiting cell growth dependent on STAT3 activity. According to the result, I have some opinions on this research, as follows:

  1. Figure 2 shows the protein expression level, not the gene expression level. Is line 290 miswrite? How many experimental repetitions is the result of the Western blotting? Please provide a quantitative histogram in each result of Western blotting. In addition, why is the value displayed under the band of GAPDH? How is this quantified?
  2. The percentage of apoptotic cells does not seem to increase significantly after treatment with acacetin for 24 or 48 h, is this not enough to support that the acacetin induces apoptosis?
  3. Why the results of p-JAK2 activity and protein expression are inconsistent in figure 4?
  4. I suggest that each immunofluorescence staining should include a scale bar.
  5. In figure 6, should it affect the phosphorylation level or the total protein level of STAT3? Will this conflict with the results in figure 1?

Author Response

(The authors gave the same response as above.)

Reviewer 3 Report

The authors presented the anti-tumor action of acacetin, focusing on its role in the phosphorylation of STST3 using DU145 prostate cancer cells. The authors employed advanced techniques such as DARTS and CETSA to identify the target of acacetin the tyrosine 705 residue. Having said that, the use of 50 uM in the cell culture study and 50 mg/kg in the animal experiment does not present the high efficacy of acacetin and the study is mostly limited to a single cell line, DU145.

Major

  • Cell line: It is recommended to use other cells besides DU145. The use of MCF10A and CCD-18Co in Figure 1 is appreciated. It is recommended to use them in response to IL6 and examine the effect of acacetin in the following assays in later figures.
  • Control: There are other STAT3 inhibitors and it is recommended to use other inhibitors as a control and compare their efficacy with that of acacetin.
  • Lines 260-262: It is not convincing based on the presented data, which show the correlation of cell viability among 3 cell lines. Besides p-STAT3, numerous differences are considered and they might be responsible for the observed difference in cell viability. It is recommended to conduct a stronger functional assay.

Minor

  • Font size: Please use the same font size throughout the text.
  • Line 39, and line 343: typo
  • Lines 158-164: Please re-write this description. The current description does not give a clear message.
  • Scale bar: Please add the scale bar to Fig. 1E, 2A, and 3B.
  • 2C: Please enlarge the labeling for X and Y axes.
  • 3C: While the gel images for ERK1/2 are shown, no description is provided in the main text.
  • 4A: The concentration of acacetin is mostly 50 uM, but it is 5 and 10 uM here. Please add the data point with 50 uM acacetin.
  • 6B (bar chart): Please add statistical significance.
  • 7D: Approximately 10% body weight change seems to indicate the toxicity of acacetin. Please add the statement on the potential toxicity of acacetin.
  • Nude mice: Please describe the strain/genotype of BALB/c mice in the materials and methods section for the animal experiment.

Author Response

(The authors gave the same response as above.)

Round 2

Reviewer 1 Report

The authors have not dissipated my concerns. I suggest to focus on the results clearly supported by data, as for binding capability and etc. 

Reviewer 2 Report

The authors responded appropriately to the comments on the original manuscript in this revised manuscript. 

Reviewer 3 Report

The authors responded appropriately to the comments on the original manuscript. Thank you.